# Delayed Emergence of Norovirus GII.17 in Finland: Foodborne Outbreaks Reported During the 2024/25 Season

**DOI:** 10.3390/v17121530

**Published:** 2025-11-21

**Authors:** Haider Al-Hello, Ruska Rimhanen-Finne, Carita Savolainen-Kopra

**Affiliations:** 1Department of Public Health, National Institute for Health and Welfare, 00300 Helsinki, Finland; ruska.rimhanen-finne@thl.fi (R.R.-F.); carita.savolainen-kopra@thl.fi (C.S.-K.); 2Faculty of Dentistry, University of Cordoba, Najaf 54001, Iraq

**Keywords:** norovirus, GII.17, retrospective analysis, phylogenetic analysis

## Abstract

During the 2024/25 season, Finland experienced a delayed emergence of norovirus genotype GII.17, which had already become prevalent in several European countries and the United States during the previous season, 2023/24. In Finland, GII.17 was confirmed in three foodborne outbreaks and five sporadic cases, marking the highest number of detections to date. These cases were geographically distributed across southern, western, and northern Finland, indicating widespread circulation. Retrospective analysis of national surveillance data from 2014 to 2025 revealed that GII.17 had been detected sporadically in earlier years but never at this scale. The emergence of GII.17 coincided with a continued presence of GII.4 [P16], the previously dominant genotype. Phylogenetic analysis showed that Finnish GII.17 strains clustered with recent international strains, suggesting introduction from abroad. The findings highlight the importance of sustained genotyping and international surveillance to detect emerging norovirus variants and inform public health preparedness in Finland.

## 1. Introduction

Human norovirus is the leading cause of acute gastroenteritis worldwide, responsible for an estimated 685 million cases annually, including 200 million among children under five [1]. The virus contributes to approximately 200,000 deaths each year, with the highest burden in low-income countries [1]. Beyond its health impact, norovirus imposes a significant economic burden, estimated at USD 60 billion globally due to healthcare costs and productivity losses [1]. In Finland, norovirus is a major cause of gastroenteritis outbreaks, particularly in institutional settings such as long-term care facilities and schools. Norovirus genotyping is conducted as a statutory part of food- and waterborne outbreak investigations [2]. Viruses in person-to-person transmissions are not routinely genotyped; instead, they are typically included in collaborative research studies. Norovirus genotyping is based on sequencing specific regions of the viral genome, particularly the capsid gene (VP1) and the RNA-dependent RNA polymerase (RdRp) gene [3]. These sequences are then compared to the typing tools implemented by the NoroNet to determine the genotype [4].

Norovirus genotype GII.17 has recently gained attention due to its rapid emergence and spread in several regions, occasionally surpassing the historically dominant GII.4 strains. Unlike earlier genotypes, GII.17 exhibits distinct amino acid substitutions in the capsid protein, particularly in the P2 subdomain, which influence its antigenic properties and receptor binding affinity. These changes may enhance immune evasion and broaden host susceptibility [5,6].

In Finland, laboratory-confirmed norovirus infections are mandatorily reported to the Finnish Infectious Diseases Register (FIDR) as part of a surveillance system coordinated by the Finnish Institute for Health and Welfare (THL). The reporting is based on nucleic acid detection methods, primarily RT-PCR, which identify human norovirus genogroups GI and GII from stool specimens. The analysis of national norovirus surveillance data from the Finnish Infectious Disease Registry (FIDR) between 2014 and 2025 reveals a biennial norovirus notification pattern with higher case numbers observed every other year (Figure 1). In the three seasons during and following the COVID-19 pandemic (2021/22, 2022/23 and 2023/24), the number of norovirus notifications resembled the levels typically seen during pre-pandemic norovirus epidemic years (2016/17 and 2018/19).

During the 2023/24 season, Finland participated in enhanced norovirus surveillance alongside other European countries and the USA [7]. A notable increase in GII.17 norovirus outbreaks and sporadic cases occurred in Austria, Germany, France, Ireland, the Netherlands, England, and the United States. This rise in GII.17 prevalence coincided with a decline in the previously dominant GII.4 genotype, with GII.17 accounting for 17% to 64% of all GII detections in the affected countries. In Finland, however, no GII.17 noroviruses were detected.

## 2. Materials and Methods

### 2.1. Surveillance and Case Reporting

Norovirus infections in Finland are mandatorily reported to the National Infectious Diseases Register (NIDR), coordinated by the Finnish Institute for Health and Welfare (THL). Reporting is based on nucleic acid detection methods, primarily reverse-transcription polymerase chain reaction (RT-PCR), which identify norovirus genogroups GI and GII from stool specimens.

### 2.2. Outbreak Surveillance

In Finland, the municipal food- and waterborne outbreak investigation groups notify suspected norovirus outbreaks to the Finnish Food and Waterborne Outbreak Registry (FWOR) [2]. We reviewed the FWOR notifications from the 2014/15 to 2024/25 seasons, in order to better understand the emergence of GII.17 in Finland.

### 2.3. Sample Selection and Genotyping

Stool specimens from outbreak-related and sporadic cases were collected and submitted for genotyping. Sporadic cases were selected from diagnostic laboratories across Finland to monitor the prevalence of noroviruses including GII.17 outside of FWOs. NoV RNA was extracted using the QIAamp Viral RNA Mini QIAcube Kit (Qiagen, Hilden, Germany) or Chemagic Viral DNA/RNA 300 Kit H96 (PerkinElmer, Inc., Waltham, MA, USA). Norovirus genotyping was conducted by sequencing regions of the viral genome from the capsid gene (VP1) and the RNA-dependent RNA polymerase (RdRp) gene using a one-step RT-PCR kit (Qiagen) according to Vinjé et al. (2004) [3]. Sequences were analyzed using Geneious 6.0.5 software [8] and Sequencher version 5.4.6 DNA sequence analysis software (Gene Codes Corporation, Ann Arbor, MI, USA). The sequences were then compared with other isolates using the NoroNet typing tool (accessed on 18 November 2025 https://mpf.rivm.nl/mpf/norovirus/typingtool/) to determine the genotype.

### 2.4. Phylogenetic Analysis

Phylogenetic trees were constructed using MEGA 12 software [9], applying the Tamura–Nei model [10] with 1000 bootstrap replications. Separate trees were generated for the capsid (VP1) and polymerase (RdRp) regions to assess genetic relationships between Finnish GII.17 strains and global reference sequences. The analysis included sequences from outbreaks and sporadic cases detected during the 2024/25 season. Finnish sequences from the preceding season were also included. All sequences have been uploaded to GenBank under accession numbers PV973171–PV973223.

### 2.5. Ethical Considerations

All data used in this study were collected as part of statutory public health surveillance and outbreak investigations. No personal identifiers were included, and ethical approval was not required for this analysis.

## 3. Results

While GII.17 was first detected in Finland in the 2015/16 season (Table 1), it remained rare and sporadic in subsequent years, with isolated detections in both outbreak-related and sporadic cases. The sporadic cases were selected to monitor the prevalence of the GII.17 strain in non-FWOs and requested from diagnostic laboratories across Finland.

During the 2024/25 season, FWOR contained 98 outbreak notifications, of which 41 (42%) were suspected to be caused by norovirus (Table 1). Noroviruses from four outbreaks were genotyped, and GII.17 was confirmed in three of them, with an estimated 164 cases. The suspected sources were dining outside the home and person-to-person transmission. Of the twelve sporadic cases, five were genotyped as GII.17. The outbreaks occurred in southern, western, and northern Finland, while the sporadic cases were reported in the south and west. This suggests that GII.17 is circulating across the entire country.

In Finland, GII.17 norovirus has been detected in both high-incidence and low-incidence years, indicating that its circulation is not strictly dependent on overall epidemic intensity. Despite the sharp decline in total human norovirus cases during the pandemic years (2019/20 and 2020/21), GII.17 re-emerged post-pandemic and was most frequently detected in the 2024/25 season, in both outbreaks and sporadic cases.

Finnish isolates from the 2024/25 season are highlighted in red, while other Finnish isolates are shown in blue. Maximum-likelihood phylogenetic trees were computed within MEGA 12 [9] using Tamura–Nei [10] with 1000 bootstrap replications. The Finnish sequence accession numbers are as follows: PV973171–PV973223; more details are provided in Appendix A.

The phylogenetic tree in Figure 2A based on the capsid gene (VP1) of GII.17 norovirus strains illustrates the genetic relationships between Finnish GII.17 sequences and global reference strains (Figure 2A). Sequences from Finland, including those recently found, cluster closely with strains from recent years (2023–2025), suggesting that the currently circulating GII.17 strains in Finland are part of the globally emerging lineage that has evolved from earlier variants first detected around 2015 and 2017.

The phylogenetic tree of the polymerase region (GII.P17) is like the VP1 phylogenetic tree; Finnish sequences from 2025 also cluster with recent international strains, indicating co-evolution of the polymerase and capsid regions. The presence of closely related sequences in both trees supports the conclusion that the current GII.17 viruses circulating in Finland are genetically like those causing outbreaks in other parts of the world.

## 4. Discussion

Norovirus GII.17 emerged later in Finland than in Central Europe and Russia, where the virus had already caused widespread outbreaks during the 2023/24 season [9]. The Finnish GII.17 strains were part of the globally expanding lineage and provided molecular evidence of their delayed introduction and spread in Finland. This phenomenon is not unique; Finland has experienced delayed emergence of epidemic viral pathogens compared to Central and Southern Europe before [11,12,13]. The true emergence of GII.17, however, might be undetected, since not all norovirus outbreaks in Finland undergo molecular investigation.

During the COVID-19 pandemic (seasons 2019/20 and 2020/21), no GII.17 norovirus was identified in Finland. Reduced virus transmission can be due to pandemic control measures: social distance, restrictions on dining together and traveling, and more hand washing leading to less outbreak investigations. After the pandemic, detection of GII.17 norovirus resumed. The actual burden of GII.17 norovirus in Finland is probably underestimated since sporadic cases likely represent a chain of person-to-person transmission, especially in settings where outbreak investigations are not routinely conducted. The rise in GII.17 detections during the 2024/25 season suggests a shift in genotype dominance and highlights the dynamic nature of norovirus epidemiology. This may partly be due to the population’s lack of exposure to norovirus infections during the pandemic, which could have led to a decline in immunity [14].

Norovirus infections are typically short-lived and self-limiting, and they rarely require hospitalization. As a result, the total number of norovirus cases often remains underreported in official statistics and only a small fraction of patients seek medical care or are recorded [15,16,17]. During the 2024–2025 season, 42% of FWOR notifications were suspected norovirus outbreaks, yet genotyping was performed in only four cases. This reflects the established practice of conducting genotyping only when exact identification is needed for comparing noroviruses in patients and the suspected source. In such cases, expert consultation is often requested from the Finnish Institute for Health and Welfare (THL), particularly when a food- or waterborne source is suspected.

The recognition of delayed genotype emergence has important public health implications. Although norovirus infections are clinically similar regardless of genotype, the introduction of a novel strain such as GII.17 may lead to larger outbreaks due to a lower population immunity [18,19,20]. Norovirus is highly contagious and can spread rapidly in institutional and community settings, often before public health interventions can be fully implemented. Thus, awareness of emerging genotypes in other regions may enhance preparedness by heightening surveillance and outbreak detection, and communication with healthcare providers. The public health authorities could also consider integrating international surveillance data into national risk assessments. Still, without targeted interventions or vaccines, the impact of enhanced surveillance and risk assessments may be limited to mitigation rather than prevention. Continued monitoring and international data sharing remain essential tools in mitigating the impact of emerging norovirus strains.

This study has several limitations that should be acknowledged. First, norovirus genotyping in Finland is not routinely performed for all outbreaks or sporadic cases, which may lead to underestimation of the true prevalence and distribution of GII.17. Genotyping is typically reserved for outbreaks with suspected foodborne or waterborne sources, limiting the representativeness of the data. Second, the retrospective nature of the surveillance data may introduce reporting biases, especially in earlier years when genotyping capacity and awareness of GII.17 were lower. Third, the sporadic cases included in this study were selected from diagnostic laboratories, which may not fully capture community transmission dynamics. Additionally, the lack of clinical data and detailed exposure histories for sporadic cases restricts our ability to assess transmission routes and outbreak linkages.

## Figures and Tables

**Figure 1 viruses-17-01530-f001:**
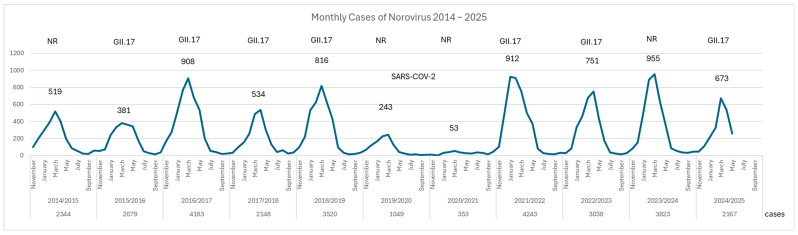
Monthly reported norovirus cases in Finland from the 2015/16 to 2024/25 seasons. This figure illustrates a biennial pattern of norovirus epidemics in Finland, with notable peaks in the 2016/17, 2018/19, 2021/22, and 2023/24 seasons. The numbers on the peaks indicate the total number of cases at each peak. The total number of cases for the entire season is mentioned below. GII.17 refers to the years in which this genotype was detected. A significant decline in cases is observed during the COVID-19 pandemic years (2019/20 and 2020/21); three years after the pandemic, norovirus case numbers were like those seen in norovirus epidemic years before the pandemic began. Notably, GII.17 norovirus has been detected in both high-incidence and low-incidence seasons, indicating its persistent circulation regardless of overall epidemic intensity. The 2024/25 season marks the highest number of GII.17 detections to date, in both outbreaks and sporadic cases. NR = no record.

**Figure 2 viruses-17-01530-f002:**
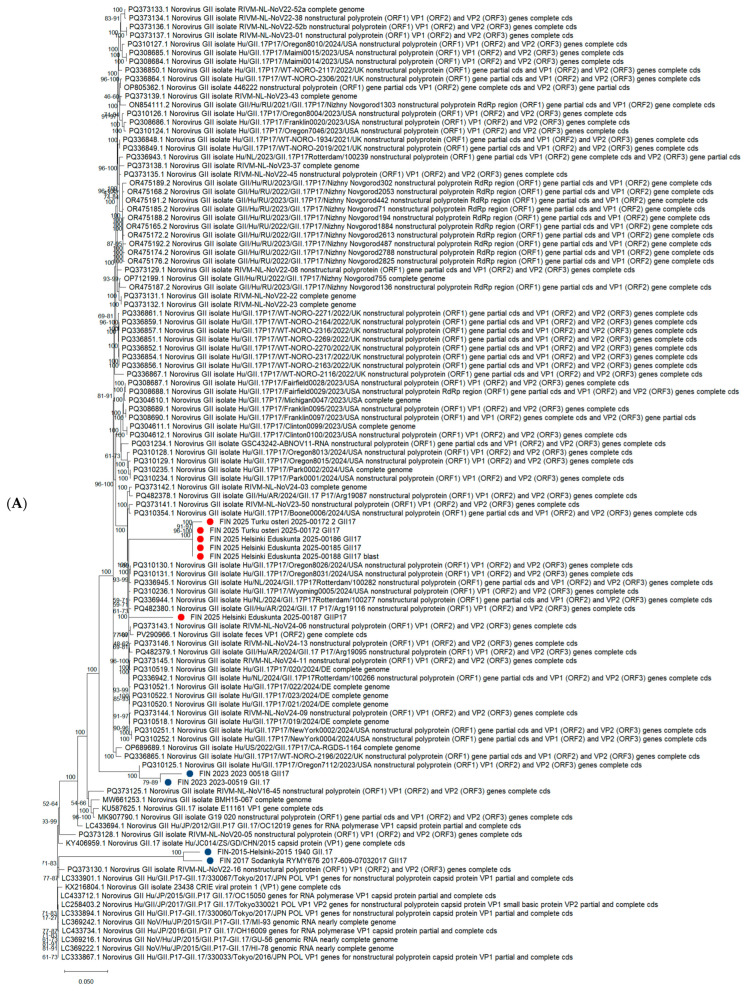
(**A**) Phylogenetic analysis of ten Finnish GII.17 VP1 sequences (200–290 bp) and 77 corresponding global sequences. (**B**) Phylogenetic analysis of 42 Finnish GII.P17 polymerase sequences (170–260 bp) and 100 corresponding global sequences.

**Table 1 viruses-17-01530-t001:** Number of food- and waterborne outbreak notifications and norovirus findings in Finland during norovirus seasons, 2014–2025.

Season	Number of Notified Outbreaks	Number of Outbreaks in Which Norovirus Is the Suspected Cause	Number of Outbreaks/ Sporadic Cases with Genotyped Noroviruses	Number of Outbreaks/Sporadic Cases with Confirmed GII.17 Finding
2014/15	67	20	10 + 48 *	NR
2015/16	80	34	24 + 20 *	2
2016/17	63	22	7 + 40 *	1 + 1 *
2017/18	80	30	21 + 37 *	4
2018/19	93	35	12 + 66 *	4 + 5 *
2019/20	72	20	4 + 27 *	NR
2020/21	58	4	1	NR
2021/22	89	26	12	1
2022/23	89	26	11 + 34 *	1 + 2 *
2023/24	108	33	9 + 23 *	NR
2024/25	98	41	4 + 12 *	3 + 5 *

Season starts in July and ends in June the subsequent year. NR stands for no record. * Sporadic case.

## Data Availability

All data generated or analyzed in this study were collected as part of statutory public health surveillance and outbreak investigations. Sequence data have been deposited in GenBank under accession numbers PV973171–PV973223. Additional details are provided in the Appendix A.

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
