# Peer review of "Delayed Emergence of Norovirus GII.17 in Finland: Foodborne Outbreaks Reported During the 2024/25 Season"

_viruses, 2025, doi:10.3390/v17121530_

Round 1

Reviewer 1 Report

Comments and Suggestions for Authors

                                               REVIEW / VIRUSES / August 2025

The authors report a retrospective analysis of National surveillance data in Finland from 2014 to 2025 revealing a delayed emergence of GII.17 norovirus strains. The phylogenetic analysis showed that these GII.17 strains were clustered with recent international norovirus strains. The study is recommendable for publication provide the authors amend according to our comments and suggestions.

Abstract

No reference is allowed to be quoted in the abstract.                                              Remove the reference 1 there.

Introduction

The referencing style is not correct in the numbering order.

For example, you can’t use on line 30 the reference 2 followed by the reference 10 on line 33 then number the reference 3 on line 36.

Materials and Methods

This section is missing. The authors should structure the sections in order as recommended by the Viruses journal in the guidelines.

Also, the ethical consideration’s part is missing in this study report.

Discussion

Line 133: Give the reference to support the statement.

References

The references 11-14 should be updated. There are, in the Norovirus literature, recent investigations which could be quoted.

Reviewer 2 Report

Comments and Suggestions for Authors

Al-Hello et al. investigated norovirus GII.17 emergence in Finland using historical case and waterborne outbreak data, and partial sequencing of positives in the VP1 and RdRp regions. The data suggest increased detection of the genotype during the 2024/25 season, occurring slightly later compared to detections in most of Europe. Interestingly, the genotype was detected even in earlier years, although this was not associated with widespread circulation. There is interest in why its outbreaks were not recorded earlier like in most of Europe. Could this be due to different subgenotypes or differences in contact patterns? The observation of a biannual epidemic pattern was also very interesting. However, I have a few minor comments and suggestions:

  1. The methods section of the manuscript is scant or even completely missing: details are needed on what qualifies to be called an outbreak, recruitment criteria for sporadic cases, sequencing approach for the RdRp and VP1, and phylogenetic methods.

  2. The phylogenetic trees need more details and can be enhanced. How many Finnish sequences are included in the trees? How many global sequeces are included. Can the Finnish sequence names be colored differently to make them easy to spot?

  3. Figure 1: The months should be indicated in English; they are currently in a different language, which makes it difficult to follow or comment further.

  4. In the introduction, the authors state that norovirus contributes to about 200,000 deaths each year, while in the discussion, they say, "Norovirus infections are typically short-lived and self-limiting, and they rarely require hospitalization." This sounds contradictory and needs clarification.

  5. In the background, it would be useful to clearly state what makes GII.17 different from previous norovirus genotypes. Are there any known amino acid changes that influence its immune evasiveness or transmissibility? What makes GII.17 unique?

  6. Cound there be subgenenotypes with GII.17.  Being an RNA virus, I would imagine its  genome signficantly evolves over time. We have a 10-year period here. Will have whole genome sequencing analysis better informed  the patterns? It will be interesting to compare the amino acid sequences VP1 sequences collected overtime.
  7. The authors need to discuss extensively the limiations of their study.

Reviewer 3 Report

Comments and Suggestions for Authors

The brief report about Norovirus genotype GII.17 by Al-Hello et al. adds important information addition to the circulation of this norovirus genotype. This is an excellent paper, it is well written, easy to read and scientifically sound. I just have a few comments on this paper.

The biggest issue in my opinion is that there is no mention of how many findings of this genotype there were, in actual numbers. It is only mentioned how many outbreaks and a few sporadic cases there were. In addition, I’m wondering if the season 2024/2025 can be called the season of emergence, since as can be seen from table 1, the genotype has been observed in all years of this study, except those where there is no records. The issue that supports the writer’s claims is that the proportion of GII.17 types as the causative agent is much greater that for example in seasons 2017-2019, when there were even more outbreaks caused by this genotype found. However, the number of typed outbreaks is very low for the season 2024/2025. Therefore, 3 out of 4 typed outbreaks showing this genotype to be the cause, could be very well be true for all other outbreaks, but also could just be a coincidence. Despite this, I still think highly of this paper and only suggest rephrasing the way this is presented.

Other detailed comments:

Figure 1: The months in this figure are in Finnish, It would be better to have everything in English. Furthermore, the numbers on top of the peaks, must be the total numbers of reported cases, not how many of the GII.17 genotypes were found. This is not clearly explained in the figure legend. From just quickly looking and reading the figure legend one could get the impression that these numbers are of this genotype. There is also no way of telling from the picture that the season 2024/2025 would be the highest number of GII.17 genotypes found.

Figure 2: It would make the figure more readable if the Finnish isolates were marked somehow, like circles or triangles etc. In addition, the season 2024/2025 could be highlighted differently from other Finnish isolates (color or shape of the marking).  

Furthermore, I’m not sure if the formatting of the paper is correct. It is divided into only two sections, “Introduction and results” and “Discussion”. However, materials and methods are included in the text and figure legends.

Yet, what is missing are the sections usually at the end: Supplementary Materials (explanation what is as a supplement), Author Contributions, Funding, Institutional Review Board Statement, Informed Consent Statement, Data Availability Statement, Acknowledgments, Conflicts of Interest, list of Abbreviations

In the supplementary file, the dates of collection were in Finnish. This might be a bit confusing.

Reviewer 4 Report

Comments and Suggestions for Authors

The Brief Report by Al-Hello et al on "Delayed emergence of norovirus GII.17 in Finland: foodborne outbreaks reported during the 2024/25 season" is generally very well written and discussed. I hace some questions and suggestions to the authors:

1- Lines 25 and 78: I would add Human noroviruses (exclusively GI and GII). Afterwards, it would be clear to readers that the article is related to these genogroups.

2- Lines 71-75: What was the case sie in there outbreaks? Were they major or minor outbreaks? Did the investigation lead to a source of infection?

3- Concerning the first phylogenetic tree (Figure 2 A):

  • I recommend placing the AN from GenBank instead of the sequence ID, plus marking the samples from this study.
  • Why only 10 sequences were included and only from 2023 to 2025 (2 from 2023, 2 from 2024, and 6 from 2025)? 
  • Remove the values of clades with a low bootsrap support index (usually everything below 70%)
  • The additive phylogram does not clearly show the formation of common clusters and, as a result, the phylogenetic relationships between the sequences included in the study, for greater representativeness, it is recommended to mark the GII.17[P17] strains included in the study, respectively;
  • Also, for greater representativeness, it is recommended specify the formation of common clusters with the compared sequences from the external group (the one that used for comparison);
  • The degree of homology and divergence between the compared sequences from the external sample is not determined; How much did the identified sequences from the Finnish outbreaks oare similar and/or differ from the sequences of the external group (the genetic distance is expressed in %);
  • It is recommended to add the length of the fragment of the nucleotide sequences RdRp and VP1 that were used in both tables;
  • It is recommended to characterize the choice of an evolutionary model when conducting a phylogenetic analysis. Why Tamura was chosen?

4- Concerning the second phylogenetic tree (Figure 2 B): Same applies, please place the AN and mark the sequences in the tree. This will allow the reader to quickly find the sequences.  Why were the 2018 strains included in this tree and excluded from the first one. The samples from this study seem to form three distinct cluster. 

5- Line 125: Please add social distancing INSTEAD of less contacts between people.

6- Line 144: This is a bit controversial as some authors report that GII.4 tend to be more severe. So, I would like to draw the attention of the authors to be less assertive. Norovirus infections is clinically similar, but the severity might differ between genotypes.

Carlson KB, Dilley A, O'Grady T, Johnson JA, Lopman B, Viscidi E. A narrative review of norovirus epidemiology, biology, and challenges to vaccine development. NPJ Vaccines. 2024 May 29;9(1):94. doi: 10.1038/s41541-024-00884-2.

5- Line 107: replace "capsid tree" by VP1 phylogenetic tree. 

Round 2

Reviewer 1 Report

Comments and Suggestions for Authors

No comments